# Tick Vaccines and Concealed versus Exposed Antigens

**DOI:** 10.3390/pathogens12030374

**Published:** 2023-02-24

**Authors:** Sandra Antunes, Ana Domingos

**Affiliations:** 1Global Health and Tropical Medicine, Institute of Hygiene and Tropical Medicine, NOVA University of Lisbon, Rua da Junqueira, 100, 1349-008 Lisboa, Portugal; 2Institute of Hygiene and Tropical Medicine, NOVA University of Lisbon, Rua da Junqueira, 100, 1349-008 Lisboa, Portugal

**Keywords:** tick control, vaccine, “exposed” antigen, “concealed” antigen

## Abstract

Anti-tick vaccines development mainly depends on the identification of suitable antigens, which ideally should have different features. These should be key molecules in tick biology, encoded by a single gene, expressed across life stages and tick tissues, capable of inducing B and T cells to promote an immunological response without allergenic, hemolytic, and toxic effects; and should not be homologous to the mammalian host. The discussion regarding this subject and the usefulness of “exposed” and “concealed” antigens was effectively explored in the publication by Nuttall et al. (2006). The present commentary intends to debate the relevance of such study in the field of tick immunological control.

## 1. Introduction

Ticks are ectoparasites capable of affecting their hosts in a dual manner as their hematophagous behavior directly affects the host, and their competency as vectors of a variety of pathogens makes these ectoparasites a major threat to animal and public health. The animal industry is familiar with ticks and tick transmitted diseases, such as anaplasmosis, ehrlichiosis, babesiosis and theileriosis, as significant economic losses are reported yearly, which are caused by animal mortality and morbidity, disease treatment and tick control. Currently, the main tick and tick-borne diseases (TTBD) control measure consists of chemical products which are directly applied to animals. These products contaminate soil, and animal derived products and are also conducive to tick resistance,. The idea of using host immunity and vaccination to combat ticks is not new. The use of naturally acquired immunity and salivary glands homogenates (known as “exposed” antigens) as sources of induced protection [1,2,3] was the first step in the endeavor to make vaccines an alternative method of tick control. Since the publication of the pioneering studies by William Trager in 1939 [3] and by Allen and Humphreys in 1979 [4], many projects have been carried out with the aim to create vaccines against ticks. In the pursuit of this goal, different paths have been followed, but the reality is that after Bm86-based vaccines were released in the 1990´s few candidates have reached the premarket stages [5]. The commercialization of these vaccines stimulated scientists to look for other antigens capable of reproducing or even surpassing the effects of the famous “Gut protein of unknown function” of the cattle tick, *Rhipicephalus* (*Boophilus*) *microplus*. This discovery focused the attention of the research community onto the idea that a “concealed” antigen could be used to elicit a robust immune response and from there the question of wheter both “exposed” and “concealed” antigens present the requisites needed to be included in a potential tick vaccine, emerged. In recent years, the usefulness of integrating one or another type of antigens in vaccines against ticks has been discussed in a diverse array of publications, in the consensus is that “concealed” antigens are defined as being usually hidden from the host immune machinery and, “exposed” antigens are defined as being secreted in tick saliva during attachment and feeding, eliciting immunological response. Hosts immunized with “exposed” antigens are boosted by permanent tick contact, while vaccination using “concealed” ones requires subsequent inoculations to ensure continuous protection [6]. After the experiments conducted by Allen and Humphreys [4], in which guinea pigs and cattle were immunized with antigens from partially fed *Dermacentor andersoni* gut and other internal organs, other reports showing that natural animal immunity against tick infestation could be enhanced using tick homogenates from whole ticks or specific organs have followed [7,8,9]. In 1988, Willadsen and Kemp, characterized a “concealed” antigen as having a key advantage in comparison to “exposed” antigens: the improbability of ticks developing immune evasion [10]. Soon after, this claim was further discussed and supported by experimental assays which included the Bm86 antigen [11,12]. Since then, several “concealed”, and “exposed” (secreted) antigens have been considered and assayed, showing both strong and weak properties toward tick infestation control. In addition to the already mentioned benefit of natural boosting, achieved using “exposed” antigens, vaccines formulated with these antigens present other relevant features: they stimulate naturally acquired resistance to tick feeding, and target different molecules and/or different stages compared with vaccines formulated with “concealed” antigens [13]. From the many experiments that have been conducted, to test potential vaccine candidates, a new idea arose which consisted of using antigens presenting characteristics offered by both the “exposed” and “concealed” types, which were named, dual-action vaccines. Using a putative tick cement protein (64P) from *Rhipicephalus appendiculatus* as an antigen, the induction of an inflammatory response at the feeding site and a simultaneous increase in antibody titers were demosntrated [14,15]. As research has advanced, other “exposed” and “concealed” antigens have shown potential as vaccine candidates and some, have been combined to improve a vaccine´s efficacy against different tick species and against tick-borne pathogens such as *Anaplasma* sp. and *Babesia* sp. [16]. Compiling relevant information regarding investigations that focused on the development of anti-tick vaccines based on both types of antigens, the 2006 publication by Nuttall et al., discussed optimal vaccine requisites and novel formulations and addressed the necessity of understanding host–parasite relationships to identify suitable candidates for vaccine development, making this manuscript a “classic paper” to researchers working in the field of TTBD immunological control.

## 2. Discovery

The path for a reduction in TTBD supported by integrated measures, to which recombinant vaccines are integral, continues to present a major constraint: the identification of protective antigens. The review by Nuttal et al. (2006), gathers experimental support for the use of both “concealed” and “exposed” antigens in the development of anti-tick vaccines, highlighting novel possibilities such as dual- action and transmission blocking vaccines (designed to block the development of parasites inside the tick, reducing tick infectivity and limiting transmission). The definition of “exposed” and “concealed” antigens and the characterization of their “mode of action”, regarding host immune response is presented. If “exposed” antigens naturally elicit a host immune response through the action of dendritic cells, which process and present these antigens to T lymphocytes, priming a cell- or antibody-mediated immune response, “concealed” ones, which are not naturally presented to host immune machinery, may provoke strong humoral immunity, producing antibodies capable of reaching and impairing their targets, which would be detrimental to tick survival [6,13]. The major problem with so-called “exposed” antigens is that during feeding, salivary glands secrete a panoply of bioactive molecules to overcome a host's haemostatic, inflammatory and immune mechanisms [17], making it improbable for a single tick antigen to induce an effective immune response. On the other hand, “concealed” antigens benefit from the element of surprise in the way that the parasite is not “prepared” for the blockage of these hidden proteins, however, the recurrent priming of the host’s immune mechanisms can be necessary. 

The classic paper also puts forward a list of assays that used the two types of antigens which were published before 2006. The example of the previously mentioned Bm86 antigen is given, and to this example, other “concealed” antigens are added, such as Vitellin, which was tested against *R. microplus* in cattle. Other “concealed” antigens were studied in different settings, for example HLS1, HLS2 and P27/30, tested against *Haemaphysalis longicornis*, Voraxin against *Amblyomma hebraeum*, and 4D8 against *Ixodes scapularis*. In parallel, “exposed” antigens such as, calreticulin, immunoglobulin-binding protein, histamine-binding protein, P29, HL 34, RIM36 and 64TRPs have been tested against different tick species, *Amblyomma americanum*, *D. variabilis* and *R. microplus*, *R. appendiculatus* and *H. longicornis,* respectively [6]. Generally, the results have been discouraging and none have reached the commercial development stage, although some of them have been tested in cattle, which is the case for 64TRPs, which was assayed against *R. appendiculatus*. At the time, recombinant versions of 64P vaccine candidates showing, not only a humoral but also a delayed-type response was given special emphasis. In this case, after immunization, tick attachment, feeding and midgut integrity were affected, causing tick death. Another advantage of this antigen is the fact that it was built with more than one conserved epitope, increasing its chances of efficacy. Progressing from the concept and findings regarding “exposed” and “concealed” antigens and dual action and transmission-blocking vaccines, the method of targeting not only the vector control but also to inhibit or reduce pathogen transmission was considered. The *R. appendiculatus* 64TRP was also tested in this regard, using *I. ricinus* infected with tick-borne encephalitis virus (TBEV) in a mouse model, showing promising results. In this study, vaccination with the recombinant protein not only impaired tick feeding and disrupted tick gut but also protected mice against a lethal infection of TBEV. The authors suggested that infection control possibly results from interactions at the level of Langerhans cells, which play a role in tick-borne TBEV transmission and are modulated by component(s) in tick saliva [18]. Given the advancements in molecular biology techniques at the time (genomics, transcriptomics and proteomics associated to, for example, RNA interference-mediated gene silencing) it was anticipated that such technologies would be key to build knowledge on the complex interactions between ticks, parasites and hosts ultimately facilitating/assisting the proposal of novel anti-TTBD vaccines. A system biology approach has been put forward, highlighting the advantages of studying the networks supporting cellular functions [16].

## 3. Impact

Sixteen years after its publication, the review entitled “Exposed and concealed antigens as vaccine targets for controlling ticks and tick-borne diseases”, by Nuttall et al., remains a milestone. This is not because it presents a ground-breaking discovery, but rather because it is a robust collection of evidence supporting the different research paths being followed in the pursuit of effective tick and tick-borne pathogens vaccines. Since then, at least 68 different “exposed” and/or “concealed” antigens have been tested in vaccination trials against different tick species aiming to achieve a reduction in tick infestation (Table 1) [19,20,21,22,23,24,25,26,27,28,29,30,31,32,33,34,35,36,37,38,39,40,41,42,43,44,45,46,47,48,49,50,51,52,53,54,55,56,57,58,59]. The attractive idea of transmission blocking vaccines has also thrived, and some studies have included the hypothesis that impairing a tick targeted antigen could also affect the pathogen life cycle occurring within its vector (Table 1). 

The involvement of the salivary glands in the fundamental process of blood feeding and in the transmission of pathogens has made it a target for research. Also, the access of host antibodies to midgut proteins has continued to stimulate interest in this tissue, but other antigens detected in ovary or eggs such as CDK10/Cyclin-dependent kinase from *I. persulcatus* [41] and vitellin-degrading enzyme [23] and yolk pro-cathepsin, from *R. microplus* [19] were also assayed in vaccination trials. However, if there was initially concern regarding the classification of antigens according to exposure to the host immune system, somewhere along the road researchers started to focus on the function and on role on tick biology rather in such classification. In recent years, what we can call the great revolution in omics approaches, has allowed the comparison of genomes, transcriptomes, proteomes and more recently metabolomes of different tick species, tissues, stages, infection and feeding statuses among others, which, allied with RNA interference technology have, from a more academic perspective, expanded knowledge on parasite biology, generating massive amounts of information on tick-host-pathogen interactions and, in a more applied view, pinpointed relevant molecules involved in fundamental tick biological processes that can be screened as protective antigens [60]. Catalogs of genes and proteins identified as differentially represented in response to a given condition are publicly available in repositories and can be screened as anti-tick vaccine candidates, with their selection being determined by the researchers’ criteria. Even though there is a growing awareness of the necessity to create models to analyze the existing data, unfortunately, these resources, which can be perceived as the first step on the road towards the discovery of suitable targets, tend to be underexplored. Reverse vaccinology, pioneered by Rino Rappuoli [61] has also been applied in anti-tick vaccine development, with the development of bioinformatic pipelines designed to identify suitable targets [54,62,63]. 

Gradually, it has also become clear that it is necessary to analyze and utilize the advantages of different antigens, epitopes, vaccine formulations and host immunological response to achieve effective protection. A synergistic effect on protection is expected when combining antigens with proved efficacy and activating different immunological mechanisms. A vaccine composed of three tick recombinant proteins, which alone conferred partial protection against *R. microplus* in confined cattle was tested in field conditions. Immunization with vitellin-degrading cysteine endopeptidase (VTDCE), *Boophilus* yolk pro-cathepsin (BYC) from *R. microplus*, and glutathione S-transferase from *H. longicornis* (GST-Hl) resulted in an increased protection level against *R. microplus* infestations in comparison to the single-antigen [64]. A promising antigen Subolesin (SUB) was combined with its ortholog Akirin, resulting in high vaccine efficacy in rabbits against *I. ricinus* and *D. reticulatus* [65] and recently, a study was published showing the potential to combine SUB with Bm86 [66]. This strategy also led to the combination of SUB with pathogen antigens to achieve a dual effect vaccine. Vaccination with SUB/ *Anaplasma marginale* Major Surface Protein 1a, resulted in a significant reduction in tick infestations in cattle and sheep, as well as a 30% reduction of *Babesia bigemina* [67]. Plus, the Q38 Subolesin/Akirin chimera containing conserved protective epitopes was also shown to be a candidate antigen to control multiple tick species infestations [68]. The peptide of the ribosomal protein P0 is another antigen presently under consideration for vaccine development. Despite being part of a conserved protein, the peptide sequence used is divergent from vertebrate hosts orthologs and is highly conserved among ticks. Peptides are small and therefore are weakly immunogenic, requiring carrier molecules to adjuvate, adding chemical stability and enhancing the immune response. The keyhole limpet haemocyanin from *Megathura crenulata* (KLH) was used together with P0 and the results showed high protection against tick infestation [37,69,70] in different tick genera. Therefore, the idea that individual Bm86 and P0 efficacies could be improved by the conjugation of both antigens [69,70,71] is being researched [37,69,70]. The great challenge of discovering an antigen suitable for anti-tick vaccine development and commercialization, remains today. However, remarkable progress in the identification and validation of candidates has been made and antigens showing promising results (particularly in cattle, due to the importance of TTBD in animal production) have supported the registration of patents. For example, alone or in alliance with Bm86, the protective antigen SUB is an antigen in which great hopes are deposited (patent US20050123554A1 and WO2014154847A1). Different studies have demonstrated positive effects against different arthropod ectoparasites and against pathogens, as reported for the mosquito borne *Plasmodium* sp. [72] and for the tick-borne *A. marginale* [38]. In 2009, results regarding the immunization of cattle with ferritin 2, against *I. ricinus* supported the request for a patent (US8168763B2). More recently, vaccination with a membrane associated protein, Aquaporin, was revealed to have high efficacy when explored in pen vaccine trials [39], allowing the registration of the US20180085443A1 patent in 2017. It is interesting to note that the latter antigens can be classified as “concealed”, supporting the concept that not only are saliva secreted (“exposed”) antigens part of an immunological equilibrium occurring between tick and host, but also that these “hidden” molecules stimulate a strong positive T cell and B cell mediated immune response. Whether an antigen is “concealed” or “exposed”, the identification of an effective antigen is only the first step in vaccine development. The work that follows its recognition as a suitable antigen for vaccine development, firstly includes the evaluation of immunogenicity, combination with adjuvants and vaccine formulations, the choice of the delivery system followed by the validation of the vaccine under controlled and in field conditions. If all the previous steps yield satisfactory results then large scale production and commercialization remain to be fulfilled. 

Immunological protection is perceived as the most sustainable tick and tick-borne diseases control method, as it circumvents acaracides drawbacks namely, the emergence of tick resistance and animal and environmental contamination. Vaccination with tick “exposed” and “concealed” antigens take advantage on either naturally acquired or artificially induced humoral immune response but more studies are needed to understand specific immunological responses that depend on a wide array of factors such as host species and/or breed, host age, immunocompetence or prior exposure to ticks. To the multifaced host immune response, tick diversity and life cycle particularities make anti tick vaccine development a complex challenge [73,74], but even though the road ahead is long, the Nuttall et al.’s article, published in 2006, is an essential read when entering the tick research world.

## Figures and Tables

**Table 1 pathogens-12-00374-t001:** Tick (Ixodidae) antigens screened in vaccination trials since 2006. The table was compiled by searching the PubMed database on November 2022, using the keywords: “tick”, “antigen” and “vaccination”. No orthologues were included and only the original vaccination study per antigen is presented.

Antigen	Accession Number	Tick Species	Antigen Descripton/Localization	Pathogen Effect	Trial Host	Year	Ref.
Yolk pro-Cathepsin	GenBank: AY966003.1	*R. microplus*	Egg		Cattle	2006	[19]
Hlim2 and Hlim3	GenBank: AB259292.1; AB252633.1	*H. longicornis*	Salivary glands		Mice	2006	[20]
Bmti n-terminal fragment	N/A (based on UniProt: P83609)	*R. microplus*	Salivary glands (serine proteinase inhibitor)		Cattle	2007	[21]
Hq02 myosin alkali light chain (MLC) proteins	GenBank: AY626788.1	*H. qinghaiensis*	Salivary glands		Sheep	2007	[22]
Vitellin-degrading enzyme	UniProt: I3VGB9	*R. microplus*	Egg (cysteine endopeptidase;		Cattle	2008	[23]
Sialostatin L2	GenBank: MK524726.1	*I. scapularis*	Salivary glands (secreted immunomodulator)		Guinea pigs	2008	[24]
Calreticulin	GenBank: AY962875	*H qinghaiensis*	Salivary glands		Sheep	2008	[25]
Metalloprotease HLMP1	GenBank: AB218891	*H. longicornis*	Salivary glands		Rabbits	2009	[26]
CHT1 chitinase	UniProt: Q8MY79	*H. longicornis*	Extracellular (secreted)		Mice	2009	[27]
Hq05	GenBank: AY626791.1	*H. qinghaiensis*	Salivary glands		Sheep	2009	[28]
Subolesin and Ubiquitin	UniProt: Q1AER5; GenBank: XP_037276798.1	*R.; R. annulatus*	Intracellular		Cattle	2010	[29]
Glutathione S-transferase	UniProt: Q6JVN0	*R. microplus*	Salivary glands		Cattle	2010	[30]
Ferritin 2 (IrFER2 and RmFER2)	GenBank: EU885951; CK190528	* I.ricinus, R. microplus, R. annulatus *	Midgut		Rabbits; Cattle	2010	[31]
5’ Nucleotidase	N/A (close to UniProt: P52307)	*R. microplus*	Ectoenzyme; Malpiguian tube		Cattle	2010	[32]
Lysosomal acid phosphatase (HL-3)	UniProt: G8C7A0	*H. longicornis*	Membrane glycoprotein (lysosome)		Rabbits	2011	[33]
Glycoprotein 97, Glycoprotein 66 and Glycoprotein 40	N/A	*H. dromedarii*	N/A		Rabbits	2011	[34]
Concinna Hc-23	GenBank: FJ425897	*H. concinna*	Intracellular (troponin like)		Rabbits	2011	[35]
Trypsin inhibitor 1-BmTI-6	GenBank: CK186726	*R. microplus*	Ovary (Kunitz-type serine protease inhibitor 6)		Cattle	2012	[36]
60S acidic ribosomal protein P0	GenBank: EU048401	*R.sanguineus*	Cell cytoplasm		Dogs	2012	[37]
SILK and Trospa	GenBank: GO496219; JK489429	*R. microplus*	Salivary glands (receptor)	*A. marginale; B. bigemina*	Cattle	2013	[38]
Aquaporin	UniProt: A0A097ITI9	*R. microplus*	Cell membrane channel		Cattle	2014	[39]
Metalloproteases Bmi-MP4	GenBank: DQ118970	*R. microplus*	Secreted in saliva		Cattle	2015	[40]
CDK10/Cyclin-dependent kinase	N/A (close to UniProt: B7P2I5)	*I. persulcatus*	Ovary		Hamster	2015	[41]
Rm76/immunoglobulin binding-protein, Rm39/cement protein, Rm239/metalloprotease and Rm180/thrombin inhibitor	ENA/EMBL-EBI: LT795750; LT795749; LT795752; LT795751	*R. microplus*	Secreted in saliva		Cattle	2017	[42]
BrBmcys2c	GenBank: AF483724	*I. persulcatus*	Salivary glands and midgut		Hamsters	2017	[43]
RmLTI and BmCG	GenBank: ACA5782 and P83606	*R. microplus*	Chimera: Bm86 + secreted serine protease inhibitors		Cattle	2018	[44]
2-Cys peroxiredoxin	N/A close to UniProt: B7Q8W6	*H. longicornis*	Midgut (peroxiredoxin)		Mice	2018	[45]
Tick Protective Antigens cocktail (heme lipoprotein, two uncharacterized secreted protein, glypican-like protein, secreted protein involved in homophilic cell adhesion, sulfate/anion exchanger, signal peptidase complex subunit 3)	GenBank: MK895447 to MK895468	*I. ricinus;D. reticulatus*	Salivary glands		Rabbits; Dogs	2019	[46]
SIFamide /myoinhibitory peptide	UniProt: B7QHG6; B7PFR9	*I. ricinus*	Synganglion (neuropeptides)	*Anaplasma* sp.	Mice; Sheep	2020	[47]
Serine protease inhibitor IrSPI and Lipocalin 1 IrLip1	GenBank: KF531922.2; MT133882	*I. ricinus*	Salivary glands	*A. phagocytophilum*	Mice; Sheep	2020	[48]
Cystatin 2a	UniProt: U3PXI8	*R. appendiculatus*	Midgut (cysteine peptidase)		Rabbits	2020	[49]
Salp14	UniProt: Q95WY7	*I. scapularis*	Salivary glands		Guinea pigs	2021	[50]
Putative glycine rich protein; Putative salivary secreted protein;RNA-binding protein	UniProt: A0A0K8R6W3; V5H126;B7PDE7	*I. ricinus*	Salivary glands		Cattle	2021	[51]
Paramyosin	GenBank: JQ517315	*H. longicornis*	Myofibrillar protein		Rabbits	2017	[52]
Lipocalin	GenBank: QGW48998	*H. longicornis*	Salivary glands		Rabbits	2021	[53]
Glutamate receptor and of a glycine-like receptor	GenBank: KF881800; KJ476181	*R. microplus*	Nervous system (synganglion); (membrane channels)		Cattle	2021	[54]
AsKunitz, AsBasicTail and As8.9kDa	GenBank: JAC23688.1; JAC23973.1; JAC23736.1	*A. sculptum*	Salivary glands		Mice	2021	[55]
Serpin RmS-17	GenBank: KC990116.1	*R. microplus*	Salivary glands		Rabbits	2022	[56]
Triosephosphate isomerase (TIM)	GenBank: MK599255	*H. longicornis*	Salivary glands		Rabbits	2022	[57]
Cement-cone protein fraction 23 kDa	N/A	*Hyalomma anatolicum; Hyalomma aegyptium*	Salivary glands		Cattle	2022	[58]
Cathepsin L-like cysteine protease	GenBank: MT905075	*H. anatolicum; Hyalomma asiaticum*	Intracellular (not tissue specific)		Rabbits	2022	[59]

## Data Availability

Not applicable.

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
