# Peer review of "Tick Vaccines and Concealed versus Exposed Antigens"

_pathogens, 2023, doi:10.3390/pathogens12030374_

Round 1

Reviewer 1 Report

Dear authors,

I have carefully read your manuscript, and found it interesting and well organized with most of the information generated in the field after the commented paper. However, several observations made directly to the document require to be corrected.

Author Response

Authors would like to acknowledge the reviewer for the corrections. Changes can be tracked on the new version of the commentary ms. We have reformulated the last paragraph as suggested. 

Author Response

Authors would like to thank the reviewer for the suggestions and corrections that have helped strengthening the commentary. As suggested by the reviewer the English was improved by an editing company.  

Line 2: The title was modified to address reviewer's concern.  

 Line 29-31: We agree with the reviewer; therefore, the Trager studies were highlighted and the reference was added.  

Line 32-33: We would like to thank the reviewer for this comment. In fact, vaccines that target ticks indirectly will affect TBD, as a reduction on tick burden will affect the transmission of pathogens, however there are cases of antigens that affect both the vector and the pathogens (64P).  

Line 42-49: The reviewer's comments were taken in account.  The phrase was replaced by two more straightforward sentences.  

Line 84-89 and 89-92: References have been included to supporting the paragraph.  

Line 139-140: After carefully reading this part of the text and, according to reviewer's observation, the authors decided to cut off the sentence as without it, the remaining text becomes clearer.  

Reviewer 3 Report

The MS comments on the possibilities to control tick infestation and tick-borne pathogen transmission using vaccines based on concealed and/or exposed tick antigens. Authors enumerate a number of papers devoted to the subject since basic publication by Nuttall et al. in 2006. The commentary tries to analyse advantages and disadvantages of vaccines based on both concealed and exposed antigens and it also touches on transmission blocking vaccines. The MS also contains an extensive table of antigens screened in vaccination trials since 2006.

One of the main problems of the MS is English. Although I am not an expert in English, I am sure that there are many mistakes that make it difficult to understand the text.

Although the MS mentions a number of works dealing with anti-tick vaccination, it does not address the host immune mechanisms involved in acquired tick resistance. It also does not explain possible principle of transmission-blocking vaccines. It does not mention new approaches to vaccination, such as mRNA or vector vaccines (Sajid et al.: mRNA vaccination induces tick resistance and prevents transmission of the Lyme disease agent. Sci Transl Med. 2021 Nov 17;13(620):eabj9827).

There are other important questions not answered (or even touched) in the MS. How can acquired rick resistance affect transmission of tick-borne pathogens? How do vaccines based on concealed antigens affect the transmission of pathogens? What is the principle of transmission-blocking vaccines?

Despite of 69 references, some are missing.

Line 115: Reference for the effect of 64TRP vaccine on the transmission of TBE is missing.

Page 3 above: The authors write: “exposed” antigens as for example, calreticulin, Immunoglobulin-binding protein, Histamine-binding protein have been tested against different tick species.  References are missing.

Line 134: Reference to Table 1 is not sufficient, references included in table should be also mentioned in the text.

Author Response

The authors would like to thank the reviewer for the revision. Lease find our answers below:

One of the main problems of the MS is English. Although I am not an expert in English, I am sure that there are many mistakes that make it difficult to understand the text. 

R: The manuscript was revised by an editing service as suggested 

Although the MS mentions a number of works dealing with anti-tick vaccination, it does not address the host immune mechanisms involved in acquired tick resistance. It also does not explain possible principle of transmission-blocking vaccines. It does not mention new approaches to vaccination, such as mRNA or vector vaccines (Sajid et al.: mRNA vaccination induces tick resistance and prevents transmission of the Lyme disease agent. Sci Transl Med. 2021 Nov 17;13(620):eabj9827). 

R: We agree with the reviewer. The explanation of the principle of transmission-blocking vaccines and the discussion of host immune mechanisms involved in acquired tick resistance or the debate on the recent technologies underline new vaccination approaches would, for sure, improve any manuscript related with tick and vaccination. However, since our objective was to produce a commentary on the Nuttall et al publication showing its importance that makes the community scientific consider it as a milestone in the field, in our point of view, we couldn’t introduce any further information than the one they published. 

There are other important questions not answered (or even touched) in the MS. How can acquired rick resistance affect transmission of tick-borne pathogens? How do vaccines based on concealed antigens affect the transmission of pathogens? What is the principle of transmission-blocking vaccines? 

R: We agree with the reviewer that these subjects are of great importance when writing about tick and tick-borne diseases control measures. If this was a scientific paper reporting and discussing data concerning to this field of work, all these answers would be, of course, included in at least the introduction and discussion chapters. However, this publication intends to be a commentary aiming to highlight what was discussed by the authors and show that this review article by Nuttall et al, published in 2006, is definitely an obligatory read for the tick research community. Nevertheless, authors gave a critical opinion about this collection of evidence acquired when searching for effective tick and parasites vaccines. 

Despite of 69 references, some are missing. 

R: Authors checked this point but as written above, the aim of this publication is not to cover all the aspects regarding tick and tick-borne diseases vaccines or control measures but to focus on what is showed in the review paper that is here commented. 

Line 115: Reference for the effect of 64TRP vaccine on the transmission of TBE is missing. 

Page 3 above: The authors write: “exposed” antigens as for example, calreticulin, Immunoglobulin-binding protein, Histamine-binding protein have been tested against different tick species.  References are missing. 

Line 134: Reference to Table 1 is not sufficient, references included in table should be also mentioned in the text. 

R: The references were added in the text. 

Round 2

Reviewer 3 Report

I agree with authors’ explanation.